# Diagnostic performance of four hepatitis-B surface antigen conformité Européenne (CE) marked and one WHO prequalified rapid diagnostic tests in Uganda

Leah Naluwagga Baliruno[1]*, Charles Drago Kato[2], Harriet Nakigozi[1], Huzaima Mujuzi[3], Emmanuel Seremba[4]

**1** Central Public Health Laboratories, Ministry of Health, Kampala, Uganda, **2** Department of Biotechnical and Diagnostic Sciences, College of Veterinary Medicine, Animal Resources and Biosecurity, Makerere University, Kampala, Uganda, **3** Department of Microbiology, UMC Victoria Hospital, Kampala, Uganda, **4** Department of Medicine, College of Health Sciences, Makerere University, Kampala, Uganda

* leabnally@gmail.com

## Abstract

### Background

Globally, over 254 million people are infected with chronic Hepatitis B (HBV). Eighty million of these reside in sub-Saharan Africa (SSA). HBV claims an estimated one million lives annually. Efforts to eradicate it from SSA have been slow, partly due to a lack of affordable, accurate screening tools. The diagnostic accuracy of the commonly used rapid diagnostic tests (RDTs) in SSA is poorly understood; hence a need to characterize the validity of five RDTs being used for HBV diagnosis using the Hepatitis B surface antigen (HBsAg) serologic marker.

### Methods

In this case-control study with 1:1 case: control matching, and a prevalence of 50%, 200 samples of residual donor blood were tested using RDTs against the reference test Fortress Diagnostic, 2016 Enzyme-Linked Immunosorbent Assay (ELISA). They were subsequently subjected to five RDT kits: the SD Bioline (Abbott Diagnostics Korea Inc.), a World Health Organization (WHO) pre-qualified test kit, and four Conformité Européenne (CE) Marked RDTs: One-Step, NOVA (Atlas Link Technology), Astracare (Astra Biotech GmbH), and Accurate (Pharmagen Uganda Ltd). Sensitivity and specificity were computed using ELISA as the reference test. The Statistical Program for Social Sciences (SPSS) was used for statistical analysis.

### Results

All five RDT brands demonstrated a sensitivity of 93% (95% CI 93%− 93%). Their specificity, however, ranged from 95% (95% CI 94.9%− 97.8%) for Astracare to 98% (95% CI 94.9%− 98.0%) for SD Bioline.

**Data availability statement:** The data underlying the results is presented in the study and is available within the paper.

**Funding:** The author(s) received no specific funding for this work.

**Competing interests:** The authors have declared that no competing interests exist.

**Abbreviations:** ART: Anti-retroviral therapy, CDC: US Centers for Disease Control and Prevention, CE: Conformité Européenne, DNA: Deoxyribonucleic Acid, EIA: Enzyme Immuno Assays, HAA: Haemo Agglutination Assays, HBsAg: Hepatitis B surface antigen, HBV: Hepatitis B Virus, HCC: Hepatocellular carcinoma, ICA: Immuno Chromatographic Assays, MOH: Ministry of Health, PCR: Polymerase Chain Reaction, POC: Point of care, PVST: Post Vaccination Serologic Testing, RDTs: Rapid Diagnostic Tests, RNA: Ribonucleic acid, UPHIA: Uganda Population-based HIV Impact Assessment.

## Conclusion

All the RDTs demonstrated good specificity and good performance characteristics. CE-marked RDTs thus present an opportunity for massive screening of the at-risk populations in the WHO-led campaign to eliminate HBV as a public health threat in Uganda and other low-resource settings by 2030.

## 1. Introduction

Hepatitis B virus (HBV) is a common cause of viral hepatitis infection with possible complications of liver failure, cirrhosis, and hepatocellular carcinoma that are responsible for an estimated mortality of 500,000–1.2 million worldwide every year [1]. Currently, the global mortality from viral hepatitis exceeds that of HIV, TB, or malaria, and is likely to exceed the toll from those three diseases combined by 2040 [2]. An estimated 1.3 million people died from viral hepatitis in 2022, from data obtained from 187 countries, and viral hepatitis is one of the communicable diseases for which mortality is increasing. Out of the 1.3 million deaths caused by hepatitis, hepatitis B caused 1.1 million deaths. The WHO African Region accounts for 63% of new hepatitis B infections. Additionally, the diagnosis coverage and end treatment coverage among all people with hepatitis B was 4% and 0.2% by the end of 2022 [3].

In its bid to eliminate HBV as a public health threat by 2030, the WHO has identified the shortage of screening tools as one of the key challenges that must be addressed for the success of the campaign [4]. This challenge is undoubtedly huge in low-resource settings, including sub-Saharan Africa. Screening and diagnosis of HBV infections is usually achieved through testing for the hepatitis B surface antigen (HBsAg). This is the first serological marker to appear in blood following infection and it becomes detectable during the incubation period [5]. Its presence signifies current infection. Several serological methods are available to detect HBsAg, including Enzyme Immuno Assays (EIA), and lateral flow or rapid diagnostic tests [6].

The EIA methods are generally used by reference laboratories and blood banks. Routine clinical testing in low-resource settings employs Rapid Diagnostic tests (RDTs) that are based on immunochromatographic principles. These tests are relatively less expensive, require minimal infrastructure and training, and provide an easy-to-interpret result within 30 minutes [7]. Various brands of RDTs demonstrated varied performance in different settings.

Varied performance characteristics were, however, recorded in East and West African populations. In Gabon, the sensitivity of the Biosynex Immunoquicks HBsAg kit was 78.0% and specificity 100% [8], and in Togo, the sensitivity of the RDT kits was 70.0% for Acon HBsAg and 95.6% for OnSite HBsAg Rapid Test-Cassette. Both kits demonstrated a specificity value of 100.0% [9]. Furthermore, in a meta study conducted, thirty studies assessed the diagnostic accuracy of 33 brands of RDTs in 23,716 individuals from 23 countries using EIA as the reference standard. The pooled sensitivity and specificity were 90.0% (95% CI: 89.1, 90.8) and 99.5% (95% CI: 99.4, 99.5), respectively, but accuracy varied widely among brands [10]. In Uganda, data

from a hospitalized population demonstrated that the Cortez rapid brand had a poor sensitivity of 43.5% and specificity of 95.8% [11].

Since 2017, the WHO has recommended one-step testing for HBV using rapid diagnostic tests in high-prevalence settings [12]. We therefore aimed to calculate the sensitivity and specificity of 4 brands of RDTs in Uganda.

## 2. Materials and methods

### 2.1. Study design

The study was carried out at the Uganda Blood Transfusion Service (UBTS), a regional branch in Kampala, Uganda, between April and May 2021. In this case-control study, five brands of hepatitis B rapid diagnostic test kits (RDTs) were evaluated for their diagnostic accuracy for viral hepatitis B. These included a WHO-prequalified SD Bioline and four CE-marked brands: One Step, NOVA, Astra care and Accurate.

### 2.2. Sample size determination and sample collection

A sensitivity of 98.9% and specificity 96.7% got from a meta-analysis study [13] was rounded up to a conservative sensitivity and specificity of 95% which was used in Buderer's formula for sample size calculation in diagnostic accuracy studies. The case control prevalence of 50%, a confidence interval of 95%, and a width of 5% was used in the Buderers' formula. Thereafter, a sample size of 99 and 34 for specificity and sensitivity, respectively, was obtained. For Buderer's formula, it states that where sensitivity and specificity are equally important to the study, the sample size for both sensitivity and specificity is determined separately, and the final sample size of the study would be the larger of these two. Given the case–control design with an assumed prevalence of 50%, the sample included 73 cases and 73 controls, making a total sample size of 146. To account for potential data loss, invalid test results, or incomplete records, an additional 10% was considered, yielding a final target sample size of approximately 160. Therefore our minimum sample size was 160 samples.

Two hundred previously collected donor blood samples available at UBTS regional branch in Kampala were purposively selected based on their HBsAg status (positive or negative for HBsAg). These were used to evaluate the diagnostic performance of five HBsAg RDTs which were particularly selected and evaluated due to their availability for routine use for HBV testing in many clinical settings in the country.

Enzyme-Linked Immunosorbent Assay (ELISA) was used as the reference standard, following the national blood bank testing algorithm. It was the only available reference standard at the time, hence this choice of reference standard. Only viable serum donor blood samples were evaluated. We included all serum samples that were sufficient in volume, < 0.75µl, and non-haemolysed. Multiple operators were available for laboratory testing.

### 2.3. Data collection and laboratory procedures

Data on socio-demographic and clinical characteristics including age, sex, area of residence, blood type, and co-infection was retrieved from archived data stored on the regional blood bank database in Kampala.

**2.3.1. Laboratory procedures.** All laboratory procedures were carried out at the UBTS laboratory in Kampala, Uganda.

**2.3.1.2 ELISA laboratory procedure.** Reference testing was done using the Fortress Diagnostic, 2016 Enzyme-Linked Immunosorbent Assay (ELISA) done according to manufacturer's instructions. It has a specificity of 99.97% and sensitivity 99.99% [14].

**2.3.1.3 Rapid diagnostic testing.** Following World Health Organization guidelines [15], HBsAg testing using rapid diagnostic tests was done on previously collected blood samples from the blood bank and tested in parallel using brands One Step Hepatitis B surface antigen Test˚ Strip (Accurate, China), NOVA test˚ China, Astracare˚ China, Accurate˚ China

and SD BIOLINE HBsAg WB, Abbott, Standard Diagnostic Inc, Korea Abbott Diagnostics Korea Inc. Samples were added to each of the test kits in accordance with the manufacturer's instructions that were clearly stated in the package inserts of each brand. Average read time for the four test kits was 10–20 minutes. The presence of HBsAg triggered the chemical reaction leading to a pink/red color observed in the test region and a similar pink/red colored band in the control region. In the absence of HBsAg, only one colored band appears in the control region and none in the test region. The total absence of color in both regions constituted an invalid test result [16]. The results were then recorded in a record form, dated and initialed by the reader. After test performance, a picture was taken of the RDT within the time period that test interpretation could be done. In case of RDT failure, the assay was repeated. When the results were discordant between the RDT test and the reference assay, those specimens with results discrepancies from the reference result were retested in duplicate using the same lot number by the same operator. The results that occurred two out of three times were recorded as the final result. When the result was discrepant again, the specimen was retested on a second lot number, if it was available. If the result on the second lot was concordant with the reference result, no further testing was performed. If the result was still discrepant from the reference results, the result was recorded as is [17].

### 2.4. Data analysis

The results generated were recorded onto worksheets and entered in excel, the collected data was then coded. The data was cleaned and checked for accuracy. The Statistical Program for Social Sciences (SPSS 20.0 for windows; SPSS Inc. Chicago, IL) was used for statistical analysis. The sensitivity and specificity of the HBsAg tests in detecting HBV infection were calculated. Pearson Chi-square test was applied for categorical variables as appropriate. P values of less than 0.05 were used to indicate statistical significance.

### 2.5. Ethical considerations

The Mulago Hospital Research and Ethics Committee (MHREC) approved the study and granted a waiver of informed consent for Protocol MHREC 2048. All methods were performed per the relevant guidelines and regulations. Administrative clearance was obtained from UBTS before the study. The research type did not require individual informed consent.

## 3. Results

### 3.1. Socio-demographic and clinical characteristics

We included 200 samples in this study. The study population consisted of young blood donors with a mean age of 26 years. The majority of participants 119 (59.5%) were male. Most of the participants were of blood group O + 83 (49.1%) followed by A + 48 (28.7%) and B + 27 (16.2%) respectively. Infection with HIV, viral hepatitis C and syphilis were uncommon in this population each at 3 (1.5%) (Table 1).

### 3.2. Sensitivity and specificity

All the study RDTs (SD Bioline, Astracare, Accurate, One Step and NOVA) demonstrated a sensitivity 93% (95% CI 93%−93%) against the ELISA platform. There was however a difference in specificity, with the SD Bioline having the highest specificity of 98% (95 CI 94.9%− 97.8%), followed by One Step at 97% (95 CI 94.9%− 97.8%). Astracare had the lowest specificity of 95% (95 CI 94.9%− 97.8%). Both Accurate and NOVA had the same specificity of 96% (95 CI 94.9%− 97.8%) (Table 2).

## 4. Discussion

Our study is among the first in Uganda to evaluate the diagnostic performance of multiple brands of RDT kits for HBsAg detection. All five brands – SD Bioline, Accurate, Astracare, One Step, and NOVA tested. All RDTs fell short of the WHO

**Table 1. Socio-demographic characteristics of blood donors.**

| Characteristics | Total N = 200 |
|---|---|
| Sex | |
| Male | 119 (67.2%) |
| Female | 58 (32.8%) |
| Age group (years) | |
| ≤20 | 62 (35%) |
| 21-30 | 66 (37.3%) |
| 31-40 | 35 (19.8%) |
| 41-50 | 12 (6.8%) |
| 51-60 | 2 (1.1%) |
| Region | |
| Central region | 118 (67.8%) |
| Northern region | 12 (6.9%) |
| West/South-Western region | 21 (12.1%) |
| Eastern region | 23 (13.2%) |
| Blood group | |
| A+ | 48 (28.7%) |
| B+ | 27 (16.2%) |
| O+ | 82 (49.1%) |
| AB+ | 8 (4.8%) |
| B- | 1 (0.6%) |
| O- | 1 (0.6%) |
| Co-infection | |
| None | 193 (96%) |
| Syphilis | 3 (1.5%) |
| HCV | 3 (1.5%) |
| HIV | 3 (1.5%) |
| HbsAg status | |
| Positive | 100 (50%) |
| Negative | 100 (50%) |

\* indicates statistical significance of a variable.

**Table 2. Sensitivity and specificity of the five brands of RDTs in the detection of HBsAg.**

| RDT brands | True positive rate | Sensitivity (Se) (95% CI) | True negative rate | Specificity (Sp) (95% CI) |
|---|---|---|---|---|
| SD BIOLINE | 93/100 | 93.0% (93.0% − 93.0%) | 98/100 | 98.0% (94.9%− 98.0%) |
| ASTRACARE | 93/100 | 93.0% (93.0% − 93.0%) | 95/100 | 95.0% (94.9%− 97.8%) |
| ACCURATE | 93/100 | 93.0% (93.0% − 93.0%) | 96/100 | 96.0% (94.9%− 97.8%) |
| ONE STEP | 93/100 | 93.0% (93.0% − 93.0%) | 97/100 | 97.0% (94.9%− 97.8%) |
| NOVA | 93/100 | 93.0% (93.0% − 93.0%) | 96/100 | 96.0% (94.9%− 97.8%) |

prequalification criteria of sensitivity set at >99%. All the kits demonstrated the same sensitivity and the specificity findings were comparable across the various kits. However the sensitivity of 93% (95% CI 93%− 93%) in this study is lower than of 100% (95% CI 98.3%− 100%) recorded by the manufacturers [18]. In earlier studies in the Ivory Coast [19],

Mongolia [13], and the Democratic Republic of Congo (DRC) [20], this kit showed a better performance with a sensitivity ranging from 98.84−100%. Unlike our study, in these populations; it met the WHO prequalification criteria of a sensitivity of ≥99% [21]. Lower sensitivities can lead to missing out on HBV cases [9]. The other brands, Astracare, Accurate, One Step and NOVA also had sensitivities of 93% (95% CI 93%− 93%) in this study, which is below the manufacturers claim of a sensitivity of >99.0%. Our findings are also lower than the 100% sensitivity recorded in the Democratic Republic of Congo [20] and in Mongolia [13]. Comparison of the performance of the Astracare and NOVA elsewhere was hindered by paucity of data in other settings. It is not clear why the diagnostic accuracy of the above kits was lower in Uganda as compared to other African countries and elsewhere. However, previous disparities in sensitivity results of rapid tests have been attributed to the existence of mutant viruses which have modified surface antigens (HBsAg) that are not readily detectable by routine immunological techniques impossible, lower HBsAg concentration and viral load that could lead to a false-negative reaction [19].

With regard to specificity, SD Bioline had a specificity of 98% (95% CI 94.9%−98.0%), the highest among the RDTs tested. This is closer to the specificity of 100% (95% CI 99.5%−100%) that was recorded by the manufacturers and comparable with the 97.1% (95% CI 96.4%–100.0%) and the 99.82% (95% CI 98.88%−100%) recorded in earlier studies in Ivory Coast [19] and DRC [20]. Similarly, the other four study kits had specificity values ranging from 95–97%, comparable to the 97.0% listed by their manufacturers. Thus, all the study RDTs met the WHO recommendation of a specificity of ≥95% [21]. For the purposes of screening and diagnosis, the sensitivity and specificity profiles of the rather inexpensive RDTs: Astracare, Accurate, One Step, and NOVA would qualify them as reasonably accurate and affordable testing platforms in resource-limited settings whose populations are highly under-tested for HBV. They would indeed be an invaluable tool to reducing the screening gap, particularly in Uganda where the population has an estimated 52% lifetime exposure to HBV, and yet 9 out of every 10 people have never tested for the disease [22]. In Uganda and some other settings where confirmatory nucleic acid testing is offered by the government or other stakeholders, false-positive results are preferable to false-negative results as positive serology often triggers repeat testing with alternative methods for case confirmation [23]. False-negative results are a potential vehicle of silent transmission and spread of the disease infection among people [19].

Generally, RDT tests with high sensitivity and negative predictive values would be more beneficial than those with high specificity and positive predictive values for routine use in poor resource settings, since negative samples from patients referred for HBV screening are rarely re-tested due to financial constraints [6]. Positive and negative predictive values are influenced by the prevalence of the disease in the population, affecting how we interpret positive and negative test results. This necessitates different approaches to follow-up testing and diagnosis confirmation depending on the epidemiological context [24].

In this study, the prevalence of HBsAg is higher in males than in females. This finding is consistent with the other African studies in South Africa [22] and Rwanda and could be attributed to males being more prone to high-risk behaviors like sexual contact, promiscuity, violence, and conflicts in which blood contact may occur [25]. The age group 21–30 years had the highest prevalence of HBsAg in this study. This is consistent with previous findings [26] and could be a reflection of the sexual transmission, as this is the most sexually active age group [27]. HBV infection being higher in younger people may also be due to their greater exposures and interaction in society as compared to children and aged persons [28]. This calls for strengthening of measures for the prevention of childhood transmission of HBV.

This study has some limitations: It was performed among blood donors who are generally healthy individuals thus results may not be generalizable to less-healthy populations such as persons living with co-infections such as HIV. Also, nucleic acid testing and sequencing were not performed, hence it is possible that occult HBV was missed.

We are also unable to add anti-Hbc core testing for the samples due to the unavailability of the testing platform. This was a limitation of the study because chronic hepatitis B was not ruled out for HbsAg-negative samples. False negatives can be reduced by adding other tests such as HBV viral load, anti-HBc testing and anti-HBs testing where possible.

Repeat testing can also be carried out after a few weeks when clinical suspicion is high, to account for the window period when HBsAg levels might be below detectable limits.

In addition, we were unable to relate a specific viral genotype or HBsAg mutant to the false-negative result. Further, we could not correlate the HBsAg quantities with RDT performance. Likewise, the study used purposive sampling of the samples thus, there was a limitation of generalizability of the results. The sample size was relatively small, and smaller sample sizes typically result in higher variability among these estimates, leading to lower precision. Additionally, the study design used 1:1 matching, meaning that the predictive values were not interpretable for this population.

Also, the biggest challenge with RDTs is that their sensitivity might not be sufficient to reliably detect low levels of infection that can occur during the window period, thus making them unsuitable for blood donor screening. Another limitation was that ELISA test results were available to the operator doing the RDT assessments it was therefore an unblinded assessment which may weaken the study quality.

## 5. Conclusions

We demonstrated that among the studied RDTs, the WHO-prequalified SD Bioline had the highest specificity among all the RDTs tested. Nevertheless, the less expensive CE-marked RDTs performed well, showing sensitivity and specificity characteristics comparable to the more expensive SD Bioline RDT. Hence, CE-marked RDTs offer affordable point-of-care tools for massive screening of at-risk populations for HBV. Their wide application can potentially contribute greatly to achieving the WHO-led goal to eliminate HBV as a public health threat by 2030.

## Supporting information

**S1 Fig. This is the S1 Fig Used rapid diagnostic tests (RDTs) showing both positive and negative strips.**
(PDF)

**S1 Appendix. This is the S1 Appendix ELISA laboratory procedure.**
(DOCX)

**S1 Dataset. This is the S1 Dataset for blood bank donor samples.**
(XLSX)

## Acknowledgments

The authors wish to thank the staff of the Uganda Blood Transfusion Services, Uganda National Laboratory Services and the College of Veterinary Medicine, Animal Resources and Biosecurity, Makerere University for their contribution to the current study. We are also grateful to the study staff and participants.

## Author contributions

**Conceptualization:** Leah Naluwagga Baliruno, Charles Drago Kato, Harriet Nakigozi, Emmanuel Seremba.

**Data curation:** Leah Naluwagga Baliruno.

**Formal analysis:** Leah Naluwagga Baliruno.

**Investigation:** Leah Naluwagga Baliruno.

**Methodology:** Huzaima Mujuzi.

**Supervision:** Charles Drago Kato, Harriet Nakigozi.

**Writing – original draft:** Leah Naluwagga Baliruno, Emmanuel Seremba.

**Writing – review & editing:** Leah Naluwagga Baliruno, Charles Drago Kato, Harriet Nakigozi, Huzaima Mujuzi, Emmanuel Seremba.

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
