## [Decision Letter · Decision Letter 0]

25 Apr 2025

Dear Dr. Baliruno,

Thank you for submitting your manuscript to PLOS ONE. After careful consideration, we feel that it has merit but does not fully meet PLOS ONE’s publication criteria as it currently stands. Therefore, we invite you to submit a revised version of the manuscript that addresses the points raised during the review process.

We look forward to receiving your revised manuscript.

Kind regards,

Seth Agyei Domfeh, PhD

Academic Editor

PLOS ONE

2. In the online submission form, you indicated that [The data underlying the results presented in the study are available from the corresponding author].

Reviewers' comments:

Reviewer's Responses to Questions

**Comments to the Author**

1. Is the manuscript technically sound, and do the data support the conclusions?

Reviewer #1: Yes

Reviewer #2: No

2. Has the statistical analysis been performed appropriately and rigorously?

Reviewer #1: Yes

Reviewer #2: No

3. Have the authors made all data underlying the findings in their manuscript fully available?

Reviewer #1: Yes

Reviewer #2: Yes

4. Is the manuscript presented in an intelligible fashion and written in standard English?

Reviewer #1: Yes

Reviewer #2: Yes

Reviewer #1: Thank you for the effort on this well written paper addressing a very important concern on limited screening tests for hepatitis in a population which has substantive exposure. I have no concern with regards to dual publication, research ethics, or publication ethics.

Please see below for few comments which you can take into consideration as you work towards a final version:

In the abstract result section and the whole of the paper there is a result on sensitivity of 93% with a CI of 93 - 93%. Please confirm this reporting version.

The cheaper RDTs here written as 'CE marked' compared with 'CE Marked' elsewhere in the document. Please consider for consistency

Introduction

Line 40 please consider deleting the word 'above'

Line 51 - 53 is this referring to only the Iranian study? Why is this part separated from the paragraph below which is addressing results from other studies?

Line 60 please consider a new paragraph

Line 68 - 70 please clarify if this is to detect acute infection

Methods and Materials

Is 200 calculation for both positives and negatives? or for each?

Is there any concern with multiple operators in the testing?

Results

In the flow chart how many participants were not included due to heamolysis for example or the flow chart starts at the ones selected for the main batch of samples?

The flow chart itself may require redoing so that it is clearer and without any overlaps

In the tables it is noted that one decimal place and 2 decimals places are used

Appendix

consider pictures of one each of the tests for clarity and submit as supplement. Add a description of the attachments

Reviewer #2: This diagnostic performance evaluation tested four HBsAg CE marked and one WHO PQ RDTs in a blood transfusion service in Uganda.

The authors evaluated 200 residual blood donor samples, with a 1:1 ratio of known positive to negative samples.

Major comments:

1. Abstract. Please state the reference ELISA and RDT manufacturers. If additional space is required, the reference to the stats program could be removed.

2. Mention in the abstract the prevalence in population (eg. 50% 1:1 case:control matching)

Introduction

3. The authors should provide a reference for this statement:

"Currently, the global mortality from viral 31 hepatitis exceeds that of HIV, TB or malaria, and is likely to exceed the toll from those three 32 diseases combined by 2040."

4. The epidemiological estimates should be updated using the WHO global hepatitis report 2024.

5. "Several immunological methods are available to detect HBsAg, including Enzyme Immuno Assays (EIA), Radio Immuno Assays (RIA), Immuno Chromatographic Assays (ICA), and Haemo Agglutination Assays (HAA)."

Both RIA and HAA are archaic methods and should not be refered to. Chemiluminescent immunoassays and EIAs are the leading methods. Immuno Chromatographic assays are commonly referred to as lateral flow or rapid diagnostic tests.

6. "In Iran, Acon, Atlas, Intec, Blue Cross, Dima and Cortez had sensitivities ranging from 97.5% -99.0 % and specificities of 97.5%- 99.2% (8). Varied performance characteristics were however recorded in East and West African populations."

This study in Iran should not be referred to in the introduction and is not clearly relevant to the current study. There are more relevant data from Africa. The statement about data from East and West African populations is more relevant but belongs as an introduction to the following paragraph. I would recommend referring to a meta-analysis or review of HBsAg RDT performance.

7. When mentioning test kits, please refer to the manufacturer and city/country of origin, not just the brand name (there are many similar brands of kits).

8. "The WHO

62 prequalified diagnostic kits are recommended internationally, however, standards for rapid

63 diagnostic test prequalification by this organization are quite stringent, making them expensive.

64 In Sub-Saharan Africa, developing countries such as Uganda, use standards for validating

65 diagnostic tools set by international regulatory bodies. These bodies include: the United States

66 Food and Drug Administration and the Conformité Européenne (CE) that are less costly as a

67 precondition for considering a diagnostic test"

I am not aware of any evidence that FDA or CE evaluation is less rigorous than the WHO PQ (or less costly). The reference provided is "ASLM. Lab culture, The ASLM Magazine. 2015;", which does not refer to a specific article (no URL or page number) or provide evidence for this assertion.

8. "However, for acute hepatitis B, RDTs may not be suitable since they can provide a false sense of security in case of negativity (14)."

This is not the case, HBsAg is positive during acute hepatitis, the reference 14 refers to national testing guidelines and not any empirical evidence to back this assertion.

9. "We therefore, aimed to calculate the sensitivity, specificity, positive and negative predictive values of 4 brands of RDTs in Uganda."

I believe you tested 5 brands?

10. The sample size formula refers to a national prevalence of 4.1% but this has not been used to calculate the sample size, instead a prevalence of 50% has been used in keeping with the study design.

11. For the Fortress diagnostic ELISA reference test, please state any available data on diagnostic/analytical sensitivity, validation data, and stringent regulatory approvals for this assay.

12. The details on methods for ELISA are not necessary, they can be moved to an appendix and just state according to manufacturers instructions.

13. Was the result of the ELISA test available to the operator doing the RDT assessments? If so, state this and list as a limitation (unblinded and not independent assessment weakens the study quality).

14. The method to retest multiple times in case of discordance with the ELISA result has resulted in an artificially (improved) assessment of test performance in relation to real conditions. This is a concern, this renders the resulting performance characteristics essentially invalid.

"Retesting was done where results were discordant, this was done to improve the accuracy of the study results"

This is inappropriate. If available, present the results before retesting multiple times ie. the first test result.

15. The flow chart is not interpretable, the lines are crossed and do not flow properly. Since the flowchart does not show any study exclusions, it is better summarised in a table (as shown in table 3).

16. Stratify table 1 by HBsAg test result, as well as showing overall characteristics.

17. A major consideration is that by taking a 1:1 case:control ratio, the sample prevalnece is 50% vs 4% nationally. Therefore all predictive values presented (PPV and NPV) are invalid, they are wildly different from a scenario when the prevalence is 4%. I would not present any predictive values. If these are to be shown, it should be shown as simulation data, weighting for a hypothetical scenario with sample prevalence of 4% (and where test performance in positive and negative samples is applied to the hypothetical scenario).

.

Reviewer #1: No

Reviewer #2: No

---

## [Author Response · Author response to Decision Letter 1]

23 Sep 2025

PONE-D-25-07381

Diagnostic Performance of Four Hepatitis-B Surface Antigen Conformité Européenne (CE) Marked and One WHO Prequalified Rapid Diagnostic Tests in Uganda

PLOS ONE

Dear Dr. Baliruno,

Thank you for submitting your manuscript to PLOS ONE. After careful consideration, we feel that it has merit but does not fully meet PLOS ONE’s publication criteria as it currently stands. Therefore, we invite you to submit a revised version of the manuscript that addresses the points raised during the review process.

We look forward to receiving your revised manuscript.

Kind regards,

Seth Agyei Domfeh, PhD

Academic Editor

PLOS ONE

2. In the online submission form, you indicated that [The data underlying the results presented in the study are available from the corresponding author].

Reviewers' comments:

Reviewer's Responses to Questions

Comments to the Author

1. Is the manuscript technically sound, and do the data support the conclusions?

Reviewer #1: Yes

Reviewer #2: No

2. Has the statistical analysis been performed appropriately and rigorously?

Reviewer #1: Yes

Reviewer #2: No

3. Have the authors made all data underlying the findings in their manuscript fully available?

Reviewer #1: Yes

Reviewer #2: Yes

4. Is the manuscript presented in an intelligible fashion and written in standard English?

Reviewer #1: Yes

Reviewer #2: Yes

5. Review Comments to the Author

Reviewer #1: Thank you for the effort on this well written paper addressing a very important concern on limited screening tests for hepatitis in a population which has substantive exposure. I have no concern with regards to dual publication, research ethics, or publication ethics.

Please see below for few comments which you can take into consideration as you work towards a final version:

Comment: In the abstract result section and the whole of the paper there is a result on sensitivity of 93% with a CI of 93 - 93%. Please confirm this reporting version.

Response: Thank you for this comment. Yes, this is the reporting version.

Comment: The cheaper RDTs here written as 'CE marked' compared with 'CE Marked' elsewhere in the document. Please consider for consistency

Response: Thank you for this comment. This has been revised in the manuscript.

Introduction

Comment: Line 40 please consider deleting the word 'above'

Response: Thank you for this comment. This has been revised in the manuscript.

Comment: Line 51 - 53 is this referring to only the Iranian study? Why is this part separated from the paragraph below which is addressing results from other studies?

Response: Thank you for this comment. The Iranian study has been replaced with references to a meta-analysis, “Diagnostic accuracy of tests to detect hepatitis B surface antigen: a systematic review of the literature and meta-analysis”

Comment: Line 60 please consider a new paragraph

Response: Thank you for this comment. This has been revised in the manuscript.

Comment: Line 68 – 70 please clarify if this is to detect acute infection

Response: Thank you for the comment. The statement has been removed because RDTs can detect both acute and chronic hepatitis B surface antigen (HbsAg).

Methods and Materials

Comment: Is 200 calculation for both positives and negatives? or for each?

Response: Thank you for this comment. The 200 calculation is for both positive and negative samples.

Comment: Is there any concern with multiple operators in the testing?

Response: Thank you for this comment. Multiple operators were employed to reduce the risk of bias.

Results

Comment: In the flow chart how many participants were not included due to heamolysis for example or the flow chart starts at the ones selected for the main batch of samples?

The flow chart itself may require redoing so that it is clearer and without any overlaps

Response: Thank you for this comment. The flow chart has been removed, as it did not add any substantial value.

Comments: In the tables it is noted that one decimal place and 2 decimals places are used

Response: Thank you for this comment. This has been revised in the manuscript to one decimal place to ensure uniformity.

Appendix

Comment: consider pictures of one each of the tests for clarity and submit as supplement. Add a description of the attachments

Response: Thank you for the comment. However, not all the pictures for each one of the tests is available. For the available picture, I have added a description.

Reviewer #2: This diagnostic performance evaluation tested four HBsAg CE marked and one WHO PQ RDTs in a blood transfusion service in Uganda.

The authors evaluated 200 residual blood donor samples, with a 1:1 ratio of known positive to negative samples.

Major comments:

1. Comment: Abstract. Please state the reference ELISA and RDT manufacturers. If additional space is required, the reference to the stats program could be removed.

Response: Thank you for this comment. This has been revised in the abstract and the manuscript.

2. Comment: Mention in the abstract the prevalence in population (e.g. 50% 1:1 case: control matching)

Response: Thank you for this comment. This has been revised accordingly in the abstract.

Introduction

3. Comment: The authors should provide a reference for this statement:

"Currently, the global mortality from viral 31 hepatitis exceeds that of HIV, TB or malaria, and is likely to exceed the toll from those three 32 diseases combined by 2040."

Response: Thank you for this comment. The reference for the above statement has been added to the manuscript.

4. Comment: The epidemiological estimates should be updated using the WHO global hepatitis report 2024.

Response: Thank you for this comment. This is kindly updated and revised using the WHO global hepatitis report 2024.

5. Comment: "Several immunological methods are available to detect HBsAg, including Enzyme Immuno Assays (EIA), Radio Immuno Assays (RIA), Immuno Chromatographic Assays (ICA), and Haemo Agglutination Assays (HAA)."

Both RIA and HAA are archaic methods and should not be refered to. Chemiluminescent immunoassays and EIAs are the leading methods. Immuno Chromatographic assays are commonly referred to as lateral flow or rapid diagnostic tests.

Response: Thank you, this is kindly noted and revised in the manuscript.

6. Comment: "In Iran, Acon, Atlas, Intec, Blue Cross, Dima and Cortez had sensitivities ranging from 97.5% -99.0 % and specificities of 97.5%- 99.2% (8). Varied performance characteristics were however, recorded in East and West African populations."

This study in Iran should not be referred to in the introduction and is not clearly relevant to the current study. There are more relevant data from Africa. The statement about data from East and West African populations is more relevant but belongs as an introduction to the following paragraph. I would recommend referring to a meta-analysis or review of HBsAg RDT performance.

Response: Thank you for this comment. The Iranian study has been replaced with references to a meta-analysis “Diagnostic accuracy of tests to detect hepatitis B surface antigen: a systematic review of the literature and meta-analysis”

7. Comment: When mentioning test kits, please refer to the manufacturer and city/country of origin, not just the brand name (there are many similar brands of kits).

Response: Thank you for the comment, this is kindly noted and revised.

8. Comment: "The WHO

62 prequalified diagnostic kits are recommended internationally, however, standards for rapid

63 diagnostic test prequalification by this organization are quite stringent, making them expensive.

64 In Sub-Saharan Africa, developing countries such as Uganda, use standards for validating

65 diagnostic tools set by international regulatory bodies. These bodies include: the United States

66 Food and Drug Administration and the Conformité Européenne (CE) that are less costly as a

67 precondition for considering a diagnostic test"

I am not aware of any evidence that FDA or CE evaluation is less rigorous than the WHO PQ (or less costly). The reference provided is "ASLM. Lab culture, The ASLM Magazine. 2015;", which does not refer to a specific article (no URL or page number) or provide evidence for this assertion.

Response: Thank you for your comment. I have updated the reference (12) alluding to the statement above, because the earlier reference was removed from the website.

8. Comment: "However, for acute hepatitis B, RDTs may not be suitable since they can provide a false sense of security in case of negativity (14)."

This is not the case, HBsAg is positive during acute hepatitis, the reference 14 refers to national testing guidelines and not any empirical evidence to back this assertion.

Response: Thank you for the comment. The statement has been removed from the manuscript following the guidance provided.

9. Comment: "We therefore, aimed to calculate the sensitivity, specificity, positive and negative predictive values of 4 brands of RDTs in Uganda."

I believe you tested 5 brands?

Response: Thank you for this comment. Yes, we tested 5 brands, and this has been corrected in the manuscript.

10. Comment: The sample size formula refers to a national prevalence of 4.1% but this has not been used to calculate the sample size, instead a prevalence of 50% has been used in keeping with the study design.

Response: Thank you for this comment. This has been kindly noted.

11. Comment: For the Fortress diagnostic ELISA reference test, please state any available data on diagnostic/analytical sensitivity, validation data, and stringent regulatory approvals for this assay.

Response: Thank you for this comment. I have added to the available details for the Fortress diagnostic ELISA reference test in the manuscript.

12. Comment: The details on methods for ELISA are not necessary, they can be moved to an appendix and just state according to manufacturer’s instructions.

Response: Thank you for comment. The details on methods for ELISA have been moved to the appendix.

13. Comment: Was the result of the ELISA test available to the operator doing the RDT assessments? If so, state this and list as a limitation (unblinded and not independent assessment weakens the study quality).

Response: The ELISA result was available to the operator doing the RDT and I have stated this as a limitation in the limitations section.

14. Comments: The method to retest multiple times in case of discordance with the ELISA result has resulted in an artificially (improved) assessment of test performance in relation to real conditions. This is a concern, this renders the resulting performance characteristics essentially invalid.

"Retesting was done where results were discordant, this was done to improve the accuracy of the study results"

This is inappropriate. If available, present the results before retesting multiple times ie. the first test result.

Response: Thank you for the comment. The wording may have implied the manipulation of the result. However, this was not the case an internal rapid diagnostic validation protocol was followed.

15. Comments: The flow chart is not interpretable, the lines are crossed and do not flow properly. Since the flowchart does not show any study exclusions, it is better summarised in a table (as shown in table 3).

Response: Thank you for the comment. The flow chart was removed as it did not add any substantial value to the study.

16. Comments: Stratify table 1 by HBsAg test result, as well as showing overall characteristics.

Response: Thank you for the comment. Overall characteristics are shown in the table. Initially the table was stratified by gender, but this

---

## [Decision Letter · Decision Letter 1]

9 Oct 2025

Dear Dr. Baliruno,

Thank you for submitting your manuscript to PLOS ONE. After careful consideration, we feel that it has merit but does not fully meet PLOS ONE’s publication criteria as it currently stands. Therefore, we invite you to submit a revised version of the manuscript that addresses the points raised during the review process.

We look forward to receiving your revised manuscript.

Kind regards,

Seth Agyei Domfeh, PhD

Academic Editor

PLOS ONE

Journal Requirements:

Reviewers' comments:

Reviewer's Responses to Questions

**Comments to the Author**

Reviewer #2: (No Response)

2. Is the manuscript technically sound, and do the data support the conclusions?

Reviewer #2: No

3. Has the statistical analysis been performed appropriately and rigorously?

Reviewer #2: No

4. Have the authors made all data underlying the findings in their manuscript fully available?

Reviewer #2: Yes

5. Is the manuscript presented in an intelligible fashion and written in standard English?

Reviewer #2: Yes

Reviewer #2: There are some unresolved issues that require revision:

1. The abstract continues to refer to 296 million people with HBV, while the introduction has been updated with Global Hepatitis report data from WHO (254 million). Please be consistent.

2. The abstract refers to SD bioline as having superior specificity relative to the other assays but the confidence intervals are entirely overlapping. I would recommend recharacterizing this as all assays having good specificity, as the finding for SD bioline may be a chance finding.

3. Introduction:

"An estimated 254 million people are living with hepatitis B. Only 13%

35 of people living with chronic hepatitis B infection had been diagnosed, presenting a big gap in

36 testing for HBV, and close to 3% had received antiviral therapy at the end of 2022. The WHO

37 African Region accounts for 63% of new hepatitis B infections (3)."

I would recommend to report the statistics for Africa from this report (4% diagnosed and 0.2% treated).

4. "Several immunological methods are available"

This isn't strictly an immunological method since we are detecting viral antigens, not antibodies. I would recharacterize as "serological methods" instead.

Here I would also mention chemiluminescent immunoassays (CLIA) as this is the most commonly used method globally.

5. "The WHO prequalified diagnostic kits are recommended

63 internationally, however, standards for rapid diagnostic test prequalification by this organization

64 are quite stringent, making them expensive."

Again, I am making the same point here: I do not think that the WHO PQ is more stringent than CE marking or FDA approval. I believe it is the opposite, fees for FDA or CE marking can exceed 200k USD whereas fees for WHO PQ are substantially lower and may be abridged if the product already has stringent regulatory approval. The WHO PQ process may be slower and requires an independent performance evaluation but to my knowledge there is no evidence it is more stringent. The cited article does not provide any evidence in support of the assertion made here.

6. Again I am making the same point here: The sample size calculation again makes reference to the Uganda local prevalence, which is not relevant, since a 50% prevalence was used in the study design. The authors used a study which showed a prevalence of 45% as the source estimate- surely this cannot be considered as the expected sensitivity. I would recalculate the sample size using a more realistic sensitivity estimate (eg. from the cited meta-analyiss) and include the prevalence of 50% as per the study design, as the current sample size calculation is not credible.

7. Again I am making the same point here: again, the study design refers to retesting if there was discrepancy, and then taking only the final result from 3 tests to evaluate at the final result in the event of discrepancy.

At the risk of repeating myself, this is not an appropriate study design since it artificially inflates the diagnostic performance. Instead, take the result from the first RDT result only (disregard the outcome of repeated testing) and use this to evaluate performance. You may later specify the result of retesting in a separate section, but do not include this in the diagnostic test evaluation.

8. Is there any room to state what is the HBsAg prevalence in blood donors locally?

9. For the 7 samples which were negative with all 5 RDTs, was there a systematic difference observed in S/CO in the ELISA result eg. close to 1? Were ELISA results checked in duplicate?

10. Discussion:

"SD Bioline, the WHO prequalified kit, had the best diagnostic performance" Again see point above about the lack of statistical difference in specificity for this test vs the others.

11. "False-negative results are a potential vehicle of silent transmit

186 ssion and spread of disease infection among people (21)."

Check spelling of "transmission" and grammar of "disease infection"

12. Add to the limitations that the study design used 1:1 matching meaning that the predictive values were not interpretable for this population.

13. I would add to the interpretation that the RDTs were found not to be suitable for a blood donor screening population as the sensitivity was insufficient.

.

Reviewer #2: No

---

## [Author Response · Author response to Decision Letter 2]

23 Jan 2026

PONE-D-25-07381R1

Diagnostic Performance of Four Hepatitis-B Surface Antigen Conformité Européenne (CE) Marked and One WHO Prequalified Rapid Diagnostic Tests in Uganda

PLOS ONE

Dear Dr. Baliruno,

Thank you for submitting your manuscript to PLOS ONE. After careful consideration, we feel that it has merit but does not fully meet PLOS ONE’s publication criteria as it currently stands. Therefore, we invite you to submit a revised version of the manuscript that addresses the points raised during the review process.

We look forward to receiving your revised manuscript.

Kind regards,

Seth Agyei Domfeh, PhD

Academic Editor

PLOS ONE

Journal Requirements:

Reviewers' comments:

Reviewer's Responses to Questions

Comments to the Author

1. If the authors have adequately addressed your comments raised in a previous round of review and you feel that this manuscript is now acceptable for publication, you may indicate that here to bypass the “Comments to the Author” section, enter your conflict of interest statement in the “Confidential to Editor” section, and submit your "Accept" recommendation.

Reviewer #2: (No Response)

2. Is the manuscript technically sound, and do the data support the conclusions?

Reviewer #2: No

3. Has the statistical analysis been performed appropriately and rigorously?

Reviewer #2: No

4. Have the authors made all data underlying the findings in their manuscript fully available?

Reviewer #2: Yes

5. Is the manuscript presented in an intelligible fashion and written in standard English?

Reviewer #2: Yes

6. Review Comments to the Author

Reviewer #2: There are some unresolved issues that require revision:

1. The abstract continues to refer to 296 million people with HBV, while the introduction has been updated with Global Hepatitis report data from WHO (254 million). Please be consistent.

Thank you for the comment. I have adjusted this accordingly.

2. The abstract refers to SD bioline as having superior specificity relative to the other assays but the confidence intervals are entirely overlapping. I would recommend recharacterizing this as all assays having good specificity, as the finding for SD bioline may be a chance finding.

Thank you, this is well noted. I have rectified this.

3. Introduction:

"An estimated 254 million people are living with hepatitis B. Only 13%

35 of people living with chronic hepatitis B infection had been diagnosed, presenting a big gap in

36 testing for HBV, and close to 3% had received antiviral therapy at the end of 2022. The WHO

37 African Region accounts for 63% of new hepatitis B infections (3)."

I would recommend to report the statistics for Africa from this report (4% diagnosed and 0.2% treated).

Thank you for this comment. The statistics have been adjusted to reflect the current HBV status.

4. "Several immunological methods are available."

This isn't strictly an immunological method since we are detecting viral antigens, not antibodies. I would recharacterize as "serological methods" instead.

Here I would also mention chemiluminescent immunoassays (CLIA) as this is the most commonly used method globally.

5. "The WHO prequalified diagnostic kits are recommended

63 internationally, however, standards for rapid diagnostic test prequalification by this organization

64 are quite stringent, making them expensive."

Again, I am making the same point here: I do not think that the WHO PQ is more stringent than CE marking or FDA approval. I believe it is the opposite, fees for FDA or CE marking can exceed 200k USD whereas fees for WHO PQ are substantially lower and may be abridged if the product already has stringent regulatory approval. The WHO PQ process may be slower and requires an independent performance evaluation but to my knowledge there is no evidence it is more stringent. The cited article does not provide any evidence in support of the assertion made here.

Thank you for this comment. This statement has been removed.

6. Again I am making the same point here: The sample size calculation again makes reference to the Uganda local prevalence, which is not relevant, since a 50% prevalence was used in the study design. The authors used a study which showed a prevalence of 45% as the source estimate- surely this cannot be considered as the expected sensitivity. I would recalculate the sample size using a more realistic sensitivity estimate (eg. from the cited meta-analyiss) and include the prevalence of 50% as per the study design, as the current sample size calculation is not credible.

Thank you for your comment. I have used the average sensitivity 98.9% and average specificity of 96.7% from a recent meta-analysis conducted and the local prevalence of hepatitis among local blood donors in Uganda. Using 50% the sample size was too low for diagnostic accuracy.

7. Again I am making the same point here: again, the study design refers to retesting if there was discrepancy, and then taking only the final result from 3 tests to evaluate at the final result in the event of discrepancy.

At the risk of repeating myself, this is not an appropriate study design since it artificially inflates the diagnostic performance. Instead, take the result from the first RDT result only (disregard the outcome of repeated testing) and use this to evaluate performance. You may later specify the result of retesting in a separate section, but do not include this in the diagnostic test evaluation.

Thank you for this comment. I have considered the first RDT test result and have removed the statement from the diagnostic test evaluation.

8. Is there any room to state what is the HBsAg prevalence in blood donors locally?

Thank you for the comment. I have added the HBsAg prevalence of blood donors locally, which ranges from 2.8% to 4.1% among blood donors in Uganda, under “sample size determination and sample collection”.

9. For the 7 samples which were negative with all 5 RDTs, was there a systematic difference observed in S/CO in the ELISA result eg, close to 1? Were ELISA results checked in duplicate?

Thank you for the comment. The ELISA results were checked in duplicate, and the S/CO was consistently below 1.0.

10. Discussion:

"SD Bioline, the WHO prequalified kit, had the best diagnostic performance." Again see point above about the lack of statistical difference in specificity for this test vs the others.

Thank you for your comment. I have removed this statement from the discussion.

11. "False-negative results are a potential vehicle of silent transmit

186 ssion and spread of disease infection among people (21)."

Check spelling of "transmission" and grammar of "disease infection"

Thank you for this comment. This has been noted and corrected accordingly.

12. Add to the limitations that the study design used 1:1 matching meaning that the predictive values were not interpretable for this population.

Thank you for this comment. I have added this to the limitations.

13. I would add to the interpretation that the RDTs were found not to be suitable for a blood donor screening population, as the sensitivity was insufficient.

Thank you for this comment. I have also as part of the limitations.

7. PLOS authors have the option to publish the peer review history of their article (what does this mean?). If published, this will include your full peer review and any attached files.

Do you want your identity to be public for this peer review? For information about this choice, including consent withdrawal, please see our Privacy Policy.

Reviewer #2: No

---

## [Decision Letter · Decision Letter 2]

10 Feb 2026

Dear Dr. Baliruno,

Thank you for submitting your manuscript to PLOS ONE. After careful consideration, we feel that it has merit but does not fully meet PLOS ONE’s publication criteria as it currently stands. Therefore, we invite you to submit a revised version of the manuscript that addresses the points raised during the review process.

We look forward to receiving your revised manuscript.

Kind regards,

Seth Agyei Domfeh, PhD

Academic Editor

PLOS One

Journal Requirements:

Reviewers' comments:

Reviewer's Responses to Questions

**Comments to the Author**

Reviewer #2: (No Response)

2. Is the manuscript technically sound, and do the data support the conclusions?

Reviewer #2: Partly

3. Has the statistical analysis been performed appropriately and rigorously?

Reviewer #2: Yes

4. Have the authors made all data underlying the findings in their manuscript fully available?

Reviewer #2: Yes

5. Is the manuscript presented in an intelligible fashion and written in standard English?

Reviewer #2: Yes

Reviewer #2: The authors have addressed most of my comments from the previous revision. However, there are two outstanding issues:

1. In section 2.2 it continues to reference a prevalence of between 2.8 to 4.1% for calculating the sample size.

This is the third time I am making the same point. The study is a case:control design, the design prevalence is 50%, you cannot use the expected population prevalence (3-4%) to estimate your sample size, when the study prevalence is 50%.

The rationale given in the comments is effectively that the result was unwanted, which is unacceptable:

"Thank you for your comment. I have used the average sensitivity 98.9% and average specificity of 96.7% from a recent meta-analysis conducted and the local prevalence of hepatitis among local blood donors in Uganda. Using 50% the sample size was too low for diagnostic accuracy."

If I apply a sample size calculation in R:

library(SampleSizeDiagnostics)

# Example assumptions

Se0 <- 0.95 # Round to more conservatives estimates

Sp0 <- 0.95

p <- 0.50 # Case control prevalence= 50%

CI <- 0.95

w <- 0.05 # Half-width of 5%

SampleSizeDiagnostics(sn = Se0, sp = Sp0, p = p, w=w, CI=CI)

Result: Sample size = 146

2. From the previous review:

"13. I would add to the interpretation that the RDTs were found not to be suitable for a blood

donor screening population, as the sensitivity was insufficient."

"Thank you for this comment. I have also as part of the limitations."

I cannot see this point added in the limitations. You mention the "window period" but the discussion does not specifically mention this important point that the observed sensitivity would be insufficient for blood donor screening according to WHO criteria, and low level HBsAg may been seen most commonly in people approaching spontaneous seroclearance/ functional cure.

.

Reviewer #2: No

---

## [Author Response · Author response to Decision Letter 3]

12 Mar 2026

Review Comments to the Author

Reviewer #2: The authors have addressed most of my comments from the previous revision. However, there are two outstanding issues:

1. In section 2.2 it continues to reference a prevalence of between 2.8 to 4.1% for calculating the sample size.

This is the third time I am making the same point. The study is a case:control design, the design prevalence is 50%, you cannot use the expected population prevalence (3-4%) to estimate your sample size, when the study prevalence is 50%.

The rationale given in the comments is effectively that the result was unwanted, which is unacceptable:

"Thank you for your comment. I have used the average sensitivity 98.9% and average specificity of 96.7% from a recent meta-analysis conducted and the local prevalence of hepatitis among local blood donors in Uganda. Using 50% the sample size was too low for diagnostic accuracy."

If I apply a sample size calculation in R:

library(SampleSizeDiagnostics)

# Example assumptions

Se0 <- 0.95 # Round to more conservatives estimates

Sp0 <- 0.95

p <- 0.50 # Case control prevalence= 50%

CI <- 0.95

w <- 0.05 # Half-width of 5%

SampleSizeDiagnostics(sn = Se0, sp = Sp0, p = p, w=w, CI=CI)

Result: Sample size = 146

Response

Thank you for the comment. I have taken into consideration the fact that this is a case-control study, and I have revised section 2.2 Sample size determination and sample collection accordingly.

2. From the previous review:

"13. I would add to the interpretation that the RDTs were found not to be suitable for a blood

donor screening population, as the sensitivity was insufficient."

"Thank you for this comment. I have also as part of the limitations."

I cannot see this point added in the limitations. You mention the "window period" but the discussion does not specifically mention this important point that the observed sensitivity would be insufficient for blood donor screening according to WHO criteria, and low level HBsAg may been seen most commonly in people approaching spontaneous seroclearance/ functional cure.

Response

Thank you for this comment. I have added this limitation in the write-up as advised.

---

## [Editor Report · Decision Letter 3]

17 Mar 2026

Diagnostic Performance of Four Hepatitis-B Surface Antigen Conformité Européenne (CE) Marked and One WHO Prequalified Rapid Diagnostic Tests in Uganda

PONE-D-25-07381R3

Dear Dr. Baliruno,

We’re pleased to inform you that your manuscript has been judged scientifically suitable for publication and will be formally accepted for publication once it meets all outstanding technical requirements.

Kind regards,

Seth Agyei Domfeh, PhD

Academic Editor

PLOS One
---

## [Editor Report · Acceptance letter]

PONE-D-25-07381R3

PLOS One

Dear Dr. Baliruno,

I'm pleased to inform you that your manuscript has been deemed suitable for publication in PLOS One. Congratulations! Your manuscript is now being handed over to our production team.

Kind regards,

on behalf of

Dr. Seth Agyei Domfeh

Academic Editor

PLOS One